# Investigating the Role of Tumor-Infiltrating Lymphocytes as Predictors of Lymph Node Metastasis in Deep Submucosal Invasive Colorectal Cancer: A Retrospective Cross-Sectional Study

**DOI:** 10.3390/cancers15215238

**Published:** 2023-10-31

**Authors:** Hirosato Tamari, Yasuhiko Kitadai, Hidehiko Takigawa, Ryo Yuge, Yuji Urabe, Fumio Shimamoto, Shiro Oka

**Affiliations:** 1Department of Gastroenterology, Hiroshima University Hospital, Hiroshima 734-8551, Japan; tamahiro@hiroshima-u.ac.jp (H.T.); hidehiko@hiroshima-u.ac.jp (H.T.); makapoo@hiroshima-u.ac.jp (R.Y.); oka4683@hiroshima-u.ac.jp (S.O.); 2Department of Health Sciences, Faculty of Human Culture and Science, Prefectural University of Hiroshima, Hiroshima 734-8558, Japan; 3Department of Gastrointestinal Endoscopy and Medicine, Hiroshima University Hospital, Hiroshima 734-8551, Japan; beyan13@hiroshima-u.ac.jp; 4Faculty of Health Sciences, Hiroshima Cosmopolitan University, Hiroshima 734-0014, Japan; simamoto@pu-hiroshima.ac.jp

**Keywords:** colorectal cancer, lymph node metastasis, tumor-infiltrating T cells, tumor microenvironment

## Abstract

**Simple Summary:**

In advanced colorectal cancer (CRC), tumor-infiltrating T cells (TILs) are reportedly associated with lymph node (LN) metastasis and prognosis. However, there are no reports focusing on early-stage CRC or examining changes in TILs according to the degree of CRC invasion. Therefore, we investigated the relationship between LN metastasis and TILs in early-stage CRC using the triple immunofluorescence method for CD4, CD8, and Foxp3. We also examined changes in TILs during invasion from early-stage CRC to advanced CRC. We found that a higher number of Foxp3+ T cells and higher Foxp3/CD4 and Foxp3/CD8 ratios at the invasive front of the tumor were associated with LN metastasis. Furthermore, tumor invasion was positively correlated with the number of TILs in CRC. This study provides new predictors of LN metastasis in T1b CRC and provides insight into the tumor microenvironment in CRC invasion.

**Abstract:**

The role of tumor-infiltrating T cells (TILs) in colorectal cancer (CRC) and their significance in early-stage CRC remain unknown. We investigated the role of TILs in early-stage CRC, particularly in deep submucosal invasive (T1b) CRC. Sixty patients with CRC (20 each with intramucosal [IM group], submucosal invasive [SM group], and advanced cancer [AD group]) were randomly selected. We examined changes in TILs with tumor invasion and the relationship between TILs and LN metastasis risk. Eighty-four patients with T1b CRC who underwent initial surgical resection with LN dissection or additional surgical resection with LN dissection after endoscopic resection were then selected. TIL phenotype and number were evaluated using triple immunofluorescence for CD4, CD8, and Foxp3. All subtypes were more numerous according to the degree of CRC invasion and more abundant at the invasive front of the tumor (IF) than in the center of the tumor (CT) in the SM and AD groups. The increased Foxp3 cells at the IF and high ratios of Foxp3/CD4 and Foxp3/CD8 positively correlated with LN metastasis. In conclusion, tumor invasion positively correlated with the number of TILs in CRC. The number and ratio of Foxp3 cells at the IF may predict LN metastasis in T1b CRC.

## 1. Introduction

Colorectal cancer (CRC) is one of the leading causes of cancer-related mortality and morbidity worldwide [1]. A complex interplay among monocytes, dendritic cells, macrophages, and T cells is evident in the tumor microenvironment of CRCs [2,3]. Various subtypes of tumor-infiltrating lymphocytes (TILs) have been delineated in CRCs, each of which plays an important role and is associated with prognosis in various types of cancer. In lung cancer, the presence of high levels of CD4+ and CD8+ T cells is a significant indicator of better prognosis [4,5,6]. In breast cancer, Foxp3+ T cells have been reported to promote tumor invasion and metastasis, and CD8 is a favorable prognostic factor [7]. In CRCs, several studies have reported associations of CD8+ and Foxp3+ T cells with prognosis [8,9,10]. However, these studies mainly focused on advanced cancer, and no studies on TILs focusing on early-stage CRC have been reported.

Early-stage CRC is defined as cancer that is confined to the intramucosa or submucosa and does not invade the muscle layer, including intramucosal (IM) CRC and submucosal invasive (SM) CRC. SM CRC includes superficial submucosal invasive (T1a) CRC (submucosal invasion depth < 1000 μm) and deep submucosal invasive (T1b) CRC (submucosal invasion depth ≥ 1000 μm). According to the Japanese Society for Cancer of the Colon and Rectum (JSCCR) 2016 guidelines [11], IM CRC or T1a is an indication for en bloc endoscopic resection. On the other hand, T1b CRC has a lymph node (LN) metastasis rate of 12.5% [12] and is an indication for intestinal resection with LN dissection. However, without the presence of pathologic risk factors for LN metastasis, such as poorly differentiated adenocarcinoma, annular cell carcinoma, mucinous carcinoma, lymphatic invasion, or budding grade 2/3 at the deepest invasive site [13,14], the LN metastasis rate can be reduced to 1.2% [15] and unnecessary surgery may be prevented. Thus, it is important to stratify the risk of LN metastasis in T1b CRC.

To examine risk factors for LN metastasis in T1b CRC, there have been detailed examinations based on histological features, but no reports examining the relationship between the tumor microenvironment and LN metastasis in detail. Moreover, no study has examined changes in the tumor microenvironment of CRC with different degrees of invasion, that is, in IM, SM, and advanced cancer (AD). Therefore, in this study, we aimed to investigate TIL characteristics according to the degree of CRC invasion and the association of TIL with the risk of LN metastasis of T1b CRC.

## 2. Materials and Methods

### 2.1. Study Population

First, 60 patients with colorectal cancer (20 with IM, 20 with SM, and 20 with AD) who were treated at Hiroshima University Hospital (Hiroshima, Japan) between December 2011 and December 2015 were randomly selected. CRC resection specimens from these 60 patients were used to examine changes in TILs according to the degree of tumor invasion.

Second, of the 102 consecutive patients with T1b CRC who underwent surgical resection with LN dissection at the beginning or additional surgical resection with LN dissection after endoscopic resection at Hiroshima University Hospital between June 2013 and August 2019, we excluded 18 patients for the following reasons: history of previous or synchronous cancer, insufficient pathological evaluation, or treatment with piecemeal resection. Finally, 84 patients were selected. Using T1b CRC resection specimens from these 84 patients, we examined the association between TIL and LN metastasis in T1b CRC. All clinical data were obtained by retrospectively reviewing patient records.

This study was conducted in accordance with the tenets of the Declaration of Helsinki and was approved by the Institutional Review Board of Hiroshima University Hospital (No. E-1518-1). Although the Ethics Committee of Hiroshima University Hospital waived the requirement for informed consent because of the use of anonymized data, informed consent was obtained from the patients using an opt-out option.

### 2.2. Histological Evaluation

All specimens were evaluated by pathologists (YK or FS) on the basis of the latest World Health Organization classification [16]. A gastrointestinal pathologist (FS) re-evaluated the specimens and made the diagnosis of T1b CRC, as described previously [17]. The researcher analyzing the quantitative data of triple immunofluorescence was blinded concerning the data. However, another researcher subsequently integrated the quantitative data with the clinical data for evaluation.

### 2.3. Assessment of the Phenotype, Number, and Distribution of TILs

The phenotype, number, and distribution of TILs were assessed using triple immunofluorescence staining for CD4, CD8, and Foxp3 (Figure 1a). The distribution of TILs was evaluated by dividing the tumor tissue into two parts, the center (CT) and the invasive front (IF) of the tumor, wherein the number and phenotype of the TILs were evaluated (Figure 1b). Specimens were examined under a fluorescence microscope (BZ-X810; Keyence, Osaka, Japan). Five high-power fields (at 200× magnification) showing substantial infiltration of TILs were selected and photographed for each sample. The numbers of CD4+, CD8+, and Foxp3+T cells were counted in each captured image using an image analyzer (BZ-H4A; Keyence, Osaka, Japan). The mean number of CD4, CD8, and Foxp3-positive T cells counted was compared relative to LN metastasis.

### 2.4. Immunofluorescence Staining

Triple immunofluorescence was performed using the Opal 4-color manual immunohistochemistry (IHC) kit (NEL810001KT; PerkinElmer, Glen Waverley, VIC, Australia). All specimens were fixed in 10% formaldehyde and embedded in paraffin according to routine procedures at the Department of Clinical Pathology, Hiroshima Universal Hospital. A 4 μm section from each specimen was cut and deparaffinized, followed by heat-induced antigen retrieval for 15 min using a microwave after-tissue pretreatment with 0.3% H_2_O_2_. The sections were then probed with an anti-CD4 antibody (1:400; ab133616; Abcam, Melbourne, VIC, Australia) overnight at 4 °C. Next, the sections were washed and incubated with secondary antibodies included in the kit for 15 min at room temperature, followed by washing and incubation at room temperature for 10 min with the Opal 570 reagent provided in the kit. After another round of heat-induced antigen retrieval, the sections were incubated with an anti-CD8 alpha antibody (1:500; ab93278; Abcam) overnight at 4 °C. The sections were then washed and incubated with secondary antibodies for 15 min at room temperature, followed by washing and incubation at room temperature for 10 min with the Opal 520 reagent. After another round of heat-induced antigen retrieval, the sections were incubated with a recombinant anti-Foxp3 antibody (1:500; ab20034; Abcam) overnight at 4 °C. They were then washed and incubated with secondary antibodies for 15 min at room temperature, followed by washing and incubation at room temperature for 10 min with the Opal 690 reagent. After washing, the sections were counterstained with DAPI (1:500) for 5 min and were mounted in mounting medium (Fluoromount; Diagnostic BioSystems, Pleasanton, CA, USA).

### 2.5. Statistical Analyses

Statistical analyses were performed using the JMP version 16.0 software (SAS Institute Inc., Cary, NC, USA). Quantitative data are presented as means ± standard deviations or percentages and were compared using Pearson’s chi-square test or Fisher’s exact test. Continuous variables were analyzed using Student’s *t*-test. Statistical significance was set at *p* < 0.05.

## 3. Results

### 3.1. Changes in the Number, Phenotype, and Distribution of TILs during CRC Progression

Patient samples were classified into three groups (IM, SM, and AD) on the basis of the depth of tumor cell infiltration. The clinicopathological characteristics of each group are presented in Table 1. In the IM group, tub/pap was the predominant histology in all cases, with no cases of positive venous invasion, positive lymphatic invasion, or LN metastasis. In the SM group, tub/pap was the predominant histology in all cases, with 18 cases of T1b CRC. Positive venous invasion was observed in two cases, and positive lymphatic invasion was observed in three. One case showed LN metastasis. In the AD group, 16 patients had predominant tub/pap histology, and 4 had predominant por/sig/muc histology. Twelve patients had positive venous invasion, and fifteen had positive lymphatic invasion. Nine patients were positive for LN metastasis, two for distant metastasis, and two for recurrence.

The CD4+, CD8+, and Foxp3+ T cells sparsely and homogeneously infiltrated the tumor tissue in the IM group, and no difference was observed in the distribution density between CT and IF (Figure 2A). In the SM group, T-cell infiltration was increased overall compared to the IM group, and hot spots were more prominent at the IF than in the CT. (Figure 2B). In the AD group, T-cell infiltration was more numerous overall than that in the SM group, and hot spots were more prominent at the IF than in the CT, as in the SM group. (Figure 2C).

In the CT and at the IF, the number of CD4+ T cells increased predominantly as the degree of infiltration progressed from M to SM and to AD (Figure 3). CD8+ and Foxp3+ T cells also predominantly increased in number with increasing invasiveness in the CT and at the IF (Figure 3).

In the IM group, there was no significant difference in the number of CD4+, CD8+, or Foxp3+ T cells between the CT and the IF. However, the number of any subtype of T cell was significantly higher at the IF than in the CT in the SM and AD groups (Figure 4).

### 3.2. Clinicopathological Characteristics of Patients with T1b CRC

We examined the association between T cells and LN metastasis in patients with T1b CRC. The clinicopathological characteristics of patients with T1b CRC are presented in Table 2. Of the 84 T1b CRC cases, 8 were positive for LN metastasis. There was no significant difference between patients with and without LN metastasis in terms of venous invasion. Lymphatic invasion was significantly more common in patients with LN metastasis than in those without LN metastasis. No significant differences were observed in the context of other risk factors for LN metastasis, such as the distance of invasion and budding grade, between the two groups.

In T1b CRC cases, CD4+, CD8+, or Fopx3+ T cells were all significantly more numerous at the IF than in the CT (Figure 5A). Next, we compared the number of TILs in patients with and without LN metastasis. In the CT, there was no significant difference in the number of CD4+, CD8+, or Fopx3+ T cells between patients with and without LN metastasis (Figure 5B). At the IF, there was no significant difference in the number of CD4+ and CD8+ T cells between the groups with and without LN metastasis. However, the number of Foxp3+ T cells at the IF was significantly higher in LN-positive cases than in LN-negative cases. Moreover, in the CT, there was no significant difference in the percentages of Foxp3/CD4 and Foxp3/CD8 between LN-positive and LN-negative cases. However, at the IF, the percentage of Foxp3/CD4 and Foxp3/CD8 was predominantly higher in LN-positive cases (Figure 5C).

However, further studies focused on LN-negative cases because some LN-negative cases had the same high percentage of Foxp3/CD4 and Foxp3/CD8 as LN-positive cases. When the presence of venous invasion was examined in the CT, there was no significant difference in the number of CD4+, CD8+, or Fopx3+ T cells between the positive venous invasion and negative venous invasion cases. However, at the IF, the number of CD4+, CD8+, or Fopx3+ T cells was predominantly higher in the positive venous invasion cases (Figure 6A). Moreover, at the IF, the percentage of Foxp3/CD4 and Foxp3/CD8 was predominantly higher in the positive venous invasion cases (Figure 6B). When the presence of lymphatic invasion was examined in the CT, there was no significant difference in the number of CD4+, CD8+, or Fopx3+ T cells between the positive lymphatic invasion and negative lymphatic invasion cases. However, at the IF, the number of CD4+, CD8+, or Fopx3+ T cells was predominantly higher in the positive lymphatic invasion cases (Figure 6C). Moreover, at the IF, the percentage of Foxp3/CD4 and Foxp3/CD8 was predominantly higher in the positive lymphatic invasion cases (Figure 6D).

## 4. Discussion

To elucidate the role of TILs in CRC, we performed triple immunostaining for CD4, CD8, and Foxp3. We found that the number of T cells significantly increased with the increasing invasiveness of CRC, including IM, SM, and AD cancer. In T1b CRC cases, a higher number and ratio of Foxp3 cells at the IF were correlated with a higher LN metastasis rate.

There has been considerable research on the tumor microenvironment, and there have been various reports on the relationship between TILs and CRC [18,19,20]. However, most of these studies focused on advanced CRC, and none have focused exclusively on early-stage CRC. Therefore, in this study, we examined how TIL changes with increasing CRC infiltration, as well as the association between T1b CRC and LN metastasis. In the study of TIL changes with CRC invasiveness, the SM group included 2 cases of T1a and 18 cases of T1b. However, since T1a is indicated for endoscopic treatment and T1b is indicated for surgery owing to the high risk of LN metastasis, we next focused on T1b and examined the risk factors for LN metastasis.

TILs play a pivotal role in tumor invasion by inducing angiogenesis and apoptosis. Furthermore, TILs activate tumor-associated macrophages through various cytokines [21]. A higher CD3/CD8 ratio in patients with rectal cancer has been correlated with a better prognosis and response to neoadjuvant chemoradiotherapy [9]. Furthermore, a higher FoxP3+ T-lymphocyte tumor infiltration score has been proven to be a favorable prognostic factor in patients with colon cancer undergoing chemotherapy or chemoimmunotherapy [22]. The location of these TILs can be divided into CT, stromal, and IF, and their prognostic relevance has been reported previously [23,24,25]. In this study, we compared the numbers of CD4, CD8, and Foxp3 cells and their changes at the IF and in the CT. We showed that TILs predominantly increase in the CT and at the IF in CRC as the tumor progresses from IM and SM to AD. The increase in TILs in proportion to the degree of infiltration represents a natural defense process of the immune system, and the accumulation of inefficient Foxp3+ T cells may suppress tumor immunity and promote tumor progression. To the best of our knowledge, this is the first study to report such findings in CRC. Future in vivo studies will provide more insight into the inefficiency of TILs in cases of LN metastases. We previously reported that macrophages predominantly increase at the IF with tumor invasion and that SM and AD cancers have hotspots at the IF [26]. In this study, similar to macrophages, TILs in SM and AD CRC also showed accumulation localization in the IF. TILs are implicated in tumor invasion, including LN metastasis and tumor recurrence [27], and the present results support this finding.

Next, we focused on early-stage T1b CRC, which has the potential to metastasize to lymph nodes. The results suggested that TILs at the IF were involved in LN metastasis. Previous studies have shown that patients with advanced CRC and high CD4/CD8 counts have high rates of LN metastasis, distant metastasis, and poor prognosis [10]. It has also been reported that LN metastasis is more common in cases with a high number and percentage of Foxp3 cells at the IF [28]. Our data showed that the LN metastasis rate was predominantly higher in cases with an increased number of Foxp3 cells at the IF and in cases with a high proportion of Foxp3/CD4 and Foxp3/CD8 at the IF, indicating that the number and proportion of Foxp3 cells at the IF are involved in LN metastasis, as has been shown by previous studies on advanced CRC [10]. An increase in Foxp3-positive T cells reportedly decreases the number and function of CD8 cells, which has antitumor effects [29,30]. Based on the findings of this study, it cannot be presumed that increased Foxp3 cell count decreased the CD4 or CD8 levels, although a higher proportion of Foxp3 in TILs and a relative decrease in CD4 and CD8 cells may have contributed to LN metastasis. Furthermore, our previous studies have shown that the number and ratio of M2 macrophages at the IF positively correlate with LN metastasis and may promote CRC progression via the epithelial–mesenchymal transition [26]. Other studies have shown that the number of Foxp3-positive T cells and macrophages correlate positively [31,32] and that Foxp3-positive T cells induce M1 macrophages to become M2 macrophages [33]. This suggests that patients with a high number of Foxp3-positive T cells at the IF may have an increased number of macrophages with M2 predominance, making the environment more prone to LN metastasis. Thus, tumor-infiltrating Foxp3 cells may affect the local immune responses and be associated with LN metastasis.

We also found several cases of T1b CRC without LN metastasis with high numbers and percentages of Foxp3 cells. A closer examination of these cases revealed that the number and percentage of Foxp3 cells at the IF were predominantly higher in cases with lymphatic or venous invasion than in those without lymphatic or venous invasion. These findings suggest that the activation of Foxp3 cells at the IF may cause lymphovascular invasion and ultimately result in LN metastasis. In fact, high Foxp3 cell counts have been reported to positively correlate with the rate of lymphovascular invasion in patients with breast cancer, and we believe that our results support this finding [34].

However, this study had some limitations. First, this was a retrospective study conducted at a single center, and the sample size was relatively small. Therefore, it is necessary to conduct prospective multicenter studies including a large number of cases in the future. Second, further research is required to validate our conclusions. Since this study examined the relationship between clinical data and propensity immunostaining, the detailed function of Fopx3 cells and the evaluation of normal tissue adjacent for the immunostaining and the H&E staining to the tumor were not investigated, and further detailed studies are needed in the future. Furthermore, we examined CD4, CD8, and Foxp3 cells with a single antibody. However, as it is now known that there are various subtypes of CD4, CD8, and Foxp3 cells, detailed studies using multiple antibodies are warranted in the future.

## 5. Conclusions

In T1b CRC, Foxp3+ T cells are implicated in CRC invasion. We believe that high numbers and high percentages of Foxp3+ T cells at the IF may be a predictor of LN metastasis.

## Figures and Tables

**Figure 1 cancers-15-05238-f001:**
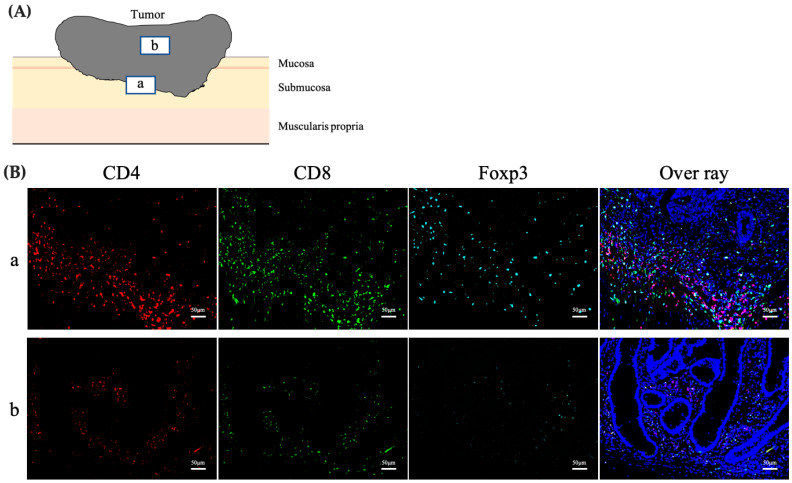
Phenotype and distribution of TILs. (**A**) Schema of the tumor. The number of TILs was counted in two different regions: (a) at the IF and (b) in the CT. (**B**) Triple immunofluorescence for CD4 (red), CD8 (green), and Foxp3 (light blue) cells. Nuclei were stained with DAPI (blue), and the overlay depicts fluorescence detected by all channels. Scale bars, 50 μm. TILs, tumor-infiltrating lymphocytes; IF, invasive front of the tumor; CT, center of the tumor; DAPI, diamidino-2-phenylindole.

**Figure 2 cancers-15-05238-f002:**
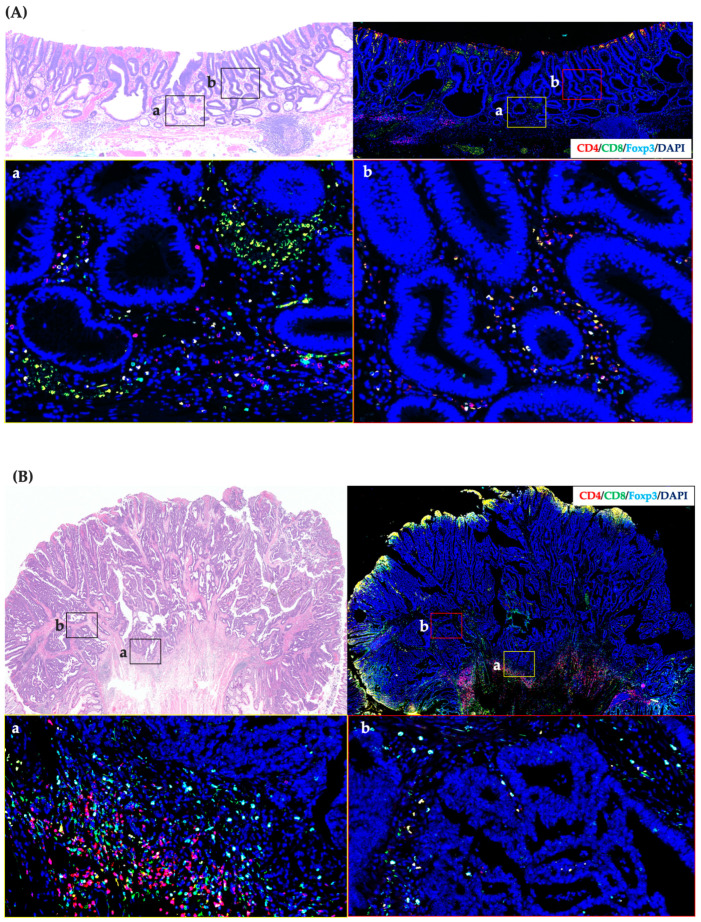
Immunofluorescence in CRC patient specimens. (**A**) Immunofluorescence image of a sigmoid CRC within the IM group. Relatively few TILs are homogeneously infiltrating the tumor stroma. (**B**) Immunofluorescence image of a sigmoid CRC within the SM group. The number of TILs in the SM group was higher than that in the IM group, and TILs were densely clustered at the invasive front. (**C**) Immunofluorescence image of an ascending CRC within the AD group. In the AD group, the number of TILs was higher than that in the SM group throughout the tumor, and the density of TILs at the invasive front was more pronounced than in the SM group. All magnified images were taken with a 200× objective lens. (a) is at the invasive front of the tumor and (b) is in the center of the tumor.

**Figure 3 cancers-15-05238-f003:**
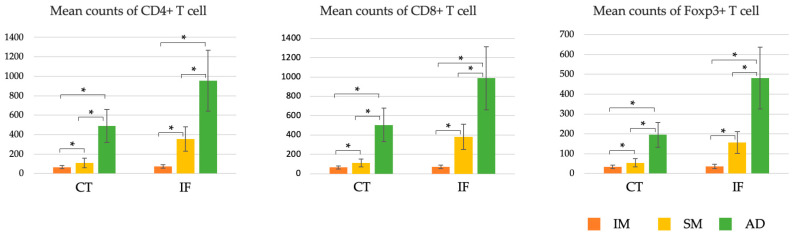
Comparison of the number of TILs in the IM, SM, and AD groups. The numbers of CD4+, CD8+, and Foxp3+ T cells increased significantly with increasing tumor invasion (M→SM→AD) at the IF and in the CT. AD, advanced cancer; CT, center of the tumor; IF, invasive front of the tumor; IM, intramucosal cancer; SM, submucosal invasive cancer; TILs, tumor-infiltrating lymphocytes. * *p* < 0.001.

**Figure 4 cancers-15-05238-f004:**
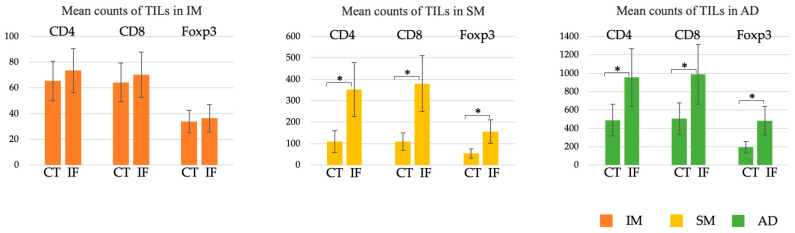
Comparison of the number of TILs in the IM, SM, and AD groups. The numbers of TILs in the CT (the left side of the bar graph) were compared to those at the IF (the right side of the bar graph). The IM group showed no significant difference in the number of CD4+, CD8+, or Fopx3+ T cells in the CT and at the IF. The SM and AD groups had significantly higher numbers of CD4, CD8, and Fopx3 cells at the IF than in the CT. AD, advanced cancer; CT, center of the tumor; IF, invasive front of the tumor; IM, intramucosal cancer; SM, submucosal invasive cancer; TILs, tumor-infiltrating lymphocytes. * *p* < 0.001.

**Figure 5 cancers-15-05238-f005:**
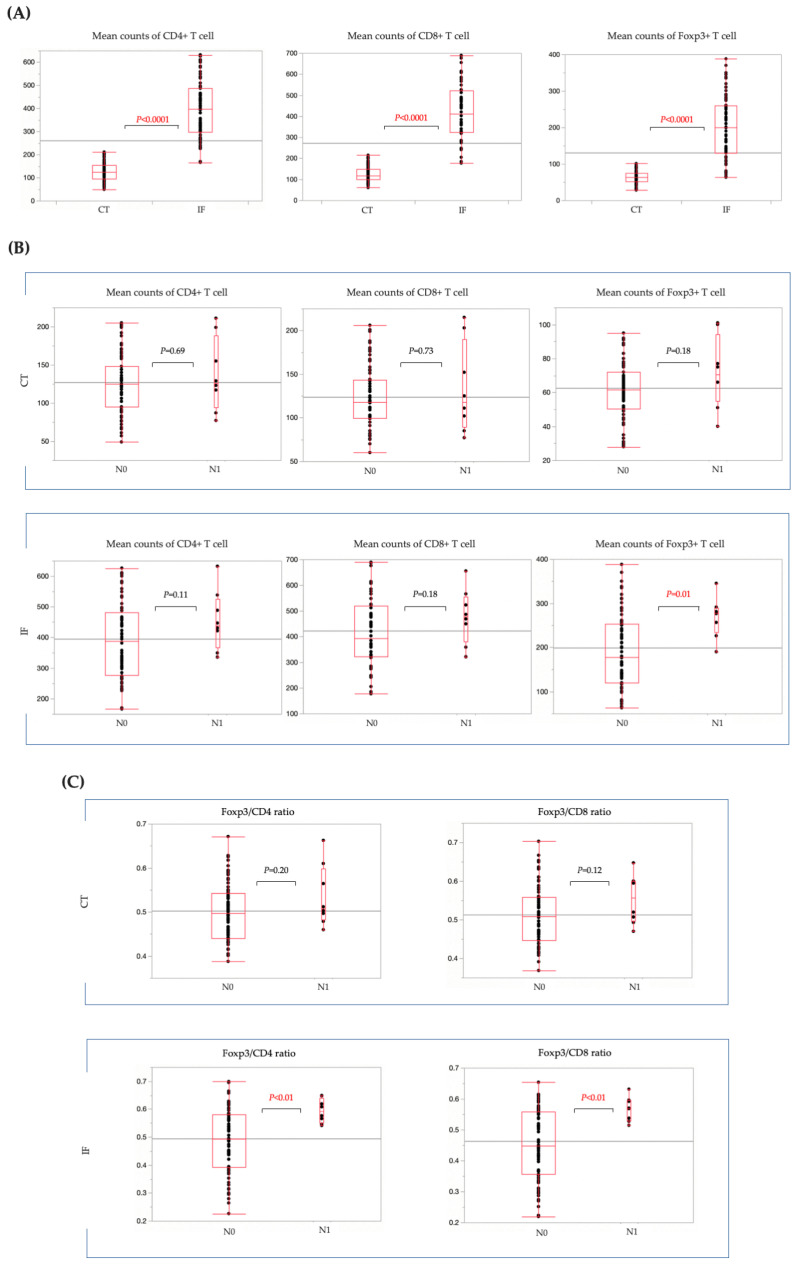
Comparison of the number of TILs in T1b CRC cases. (**A**) CD4+, CD8+, and Fopx3+ T cells were significantly more numerous at the IF than in the CT. (**B**) No significant difference was observed in terms of CD4+, CD8+, and Foxp3+ T cell counts in the CT between patients with and without LN metastasis. At the IF, there was no significant difference in CD4+ and CD8+ T cell count, although Foxp3+ T cells were significantly more numerous in the LN-positive cases. (**C**) No significant difference was observed in the ratio of Foxp3/CD4 to Foxp3/CD8 in the CT between patients with and without LN metastasis. Patients with LN metastasis had significantly higher Foxp3/CD4 and Foxp3/CD8 ratios at the IF than patients without LN metastasis. CRC, colorectal cancer; CT, center of the tumor; IF, invasive front of the tumor; N0, patients without LN metastasis; N1, patients with LN metastasis; TILs, tumor-infiltrating lymphocytes; T1b CRC, submucosal invasive colorectal cancer.

**Figure 6 cancers-15-05238-f006:**
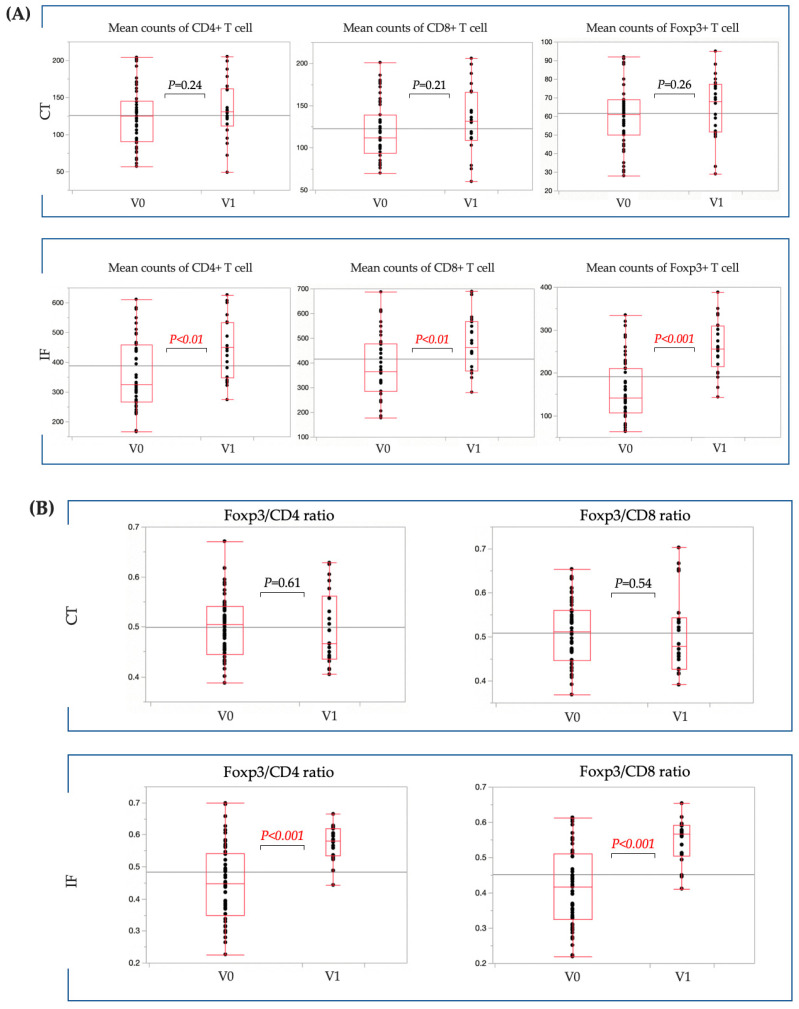
Comparison of the counts of TILs in T1b CRC without LN metastasis. (**A**) No significant difference was observed in terms of CD4+, CD8+, and Foxp3+ T cell counts in the CT between patients with and without venous invasion. Patients with venous invasion had significantly higher numbers of CD4+, CD8+, and Foxp3+ T cells at the IF than patients without venous invasion. (**B**) No significant difference was evident in the ratio of Foxp3/CD4 to Foxp3/CD8 in the CT between patients with and without venous invasion. Patients with venous invasion demonstrated significantly higher Foxp3/CD4 and Foxp3/CD8 ratios at the IF than patients without venous invasion. (**C**) No significant difference was noted in terms of CD4+, CD8+, and Foxp3+ T cell counts in the CT between patients with and without lymphatic invasion. Patients with lymphatic invasion had significantly higher numbers of CD4+, CD8+, and Foxp3+ T cell counts at the IF than patients without lymphatic invasion. (**D**) No significant difference in terms of the ratio of Foxp3/CD4 to Foxp3/CD8 in the CT was observed between patients with and without lymphatic invasion. Patients with lymphatic invasion had significantly higher Foxp3/CD4 and Foxp3/CD8 ratios at the IF than patients without lymphatic invasion. TILs, tumor-infiltrating lymphocytes; T1b CRC, submucosal invasive colorectal cancer; CT, center of the tumor; IF, invasive front of the tumor; CRC, colorectal cancer; Ly0, without lymphatic invasion; Ly1, with lymphatic invasion; V0, without venous invasion; V1, with venous invasion.

**Table 1 cancers-15-05238-t001:** Clinicopathological characteristics of patients with colorectal cancer.

Characteristics	IM	SM *	AD
	(*n* = 20)	(*n* = 20)	(*n* = 20)
Age, mean ± SD, years	65 ± 10.2	67 ± 11.3	66 ± 8.8
Sex			
Male	13 (65)	12 (59)	13 (65)
Female	7 (35)	8 (40)	7 (35)
Tumor size, mean ± SD, mm	14.7 ± 4.3	16.4 ± 5.2	28.8 ± 12.5
Localization			
Right colon	7 (35)	8 (40)	7 (35)
Left colon	6 (30)	6 (30)	8 (40)
Rectum	7 (35)	6 (30)	5 (25)
Morphology			
Protruded	12 (60)	7 (35)	20 (100)
Superficial/Depressed	8 (40)	33	0 (0)
Dominant histological type			
tub/pap	20 (100)	20 (100)	16 (80)
por/sig/muc	0 (0)	0 (0)	4 (20)
Venous invasion positive	0 (0)	2 (10)	12 (60)
Lymphatic invasion positive	0 (0)	3 (15)	15 (75)
LN metastasis positive	0 (0)	1 (5)	9 (45)
Distant metastasis positive	0 (0)	0 (0)	2 (10)
Recurrence	0 (0)	0 (0)	2 (10)
			(%)

* SM group includes 2 cases of T1a and 18 of T1b.

**Table 2 cancers-15-05238-t002:** Clinicopathological characteristics of patients with T1b CRC.

Characteristics		LN Metastasis	
	(+), *n* = 8	(−), *n* = 76	*p*-Value
Age, mean ± SD, years	68 ± 10.5	67 ± 10.2	0.79
Sex			0.62
Male	4 (50)	45 (59)	
Female	4 (50)	31 (41)	
Tumor size, mean ± SD, mm	26 ± 16.6	24 ± 10.4	0.96
Localization			0.81
Right colon	2 (25)	27 (36)	
Left colon	4 (50)	35 (46)	
Rectum	2 (25)	14 (18)	
Morphology			0.022
Protruded	3 (38)	59 (78)	
Superficial	5 (62)	17 (22)	
Treatment			0.34
Surgery alone	5 (63)	34 (45)	
Surgery after endoscopic resection	3 (37)	42 (55)	
Dominant histological type			1
tub/pap	8 (100)	76 (100)	
por/sig/muc	0 (0)	0 (0)	
SM invasion depth, mean ± SD, μm	3687 ± 1614	3745 ± 2166	0.73
Venous invasion positive	5 (63)	22 (29)	0.063
Lymphatic invasion positive	7 (88)	27 (36)	0.0036
Budding grade 2/3	6 (75)	35(46)	0.11
Distant metastasis positive	0 (0)	0 (0)	1
Recurrence	0 (0)	0 (0)	1
			(%)

## Data Availability

The data that support the findings of this study are available from the corresponding author upon reasonable request.

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
