# Peer review of "Investigating the Role of Tumor-Infiltrating Lymphocytes as Predictors of Lymph Node Metastasis in Deep Submucosal Invasive Colorectal Cancer: A Retrospective Cross-Sectional Study"

_cancers, 2023, doi:10.3390/cancers15215238_

Round 1

Reviewer 1 Report

Comments and Suggestions for Authors

Although the authors have tried to study the "role of tumor infiltrating T cells (TILs) in colorectal cancer (CRC) and their significance in CRC. However, the manuscript needs some changes to make it comprehensive and acceptable for the journal.

The comments are below.

1. The method(s)-material part needs to be elaborated.

2. No mention of the Control(adjacent tissue from healthy control) use for the phenotype/distribution of the TILs.

3. Authors have not mentioned about the Ethical Approval.

4.   Authors have not mentioned the Control(adjacent tissue from healthy control)  for the Immunostaining and have not done the H&E staining.

5. Retrospective nature of the study and use of single antibody bias needs to be elaborated.

6. English language and scientific language needs to be checked again.

Comments on the Quality of English Language

English language and scientific language needs to be checked again.

Author Response

September 25, 2023

Dear Editor:

We thank you for the thorough review of our manuscript and your thoughtful suggestions and insights. We have taken the reviewer’s comments into account and have made adjustment to the text accordingly. We have highlighted changes in the manuscript in red.

The manuscript has been rechecked, and necessary changes have been made in accordance with the reviewers’ suggestions. The responses to all comments have been prepared and given below. I look forward to working with you and the reviewers to move this manuscript closer to publication in Cancers.

Sincerely,

Yasuhiko Kitadai, PhD

Department of Health Sciences, Faculty of Human Culture and Science

Prefectural University of Hiroshima

1-1-71, Ujinahigashi, Minami-ku, Hiroshima 734-8558, Japan

Responses to the Reviewer

Our point-by-point response to the comments is as follows.

  1. The method(s)-material part needs to be elaborated.

Reply: Thank you for your comment. I have described the methods and materials in more detail as you advised. I have also described the result section in more detail.

  1. No mention of the Control(adjacent tissue from healthy control) use for the phenotype/distribution of the TILs.

Reply: Thank you for your comment. As you have mentioned, it is very important to consider the adjacent tissue of the donation target. Many prior reports have evaluated adjacent tissues of healthy controls. However, since our study focused on M carcinoma, T1 carcinoma, and advanced cancer lesions and examined TIL changes in tumor invasion, we were not able to consider adjacent tissue. However, this is a very important point and is therefore described in the limitation section.

  1. Authors have not mentioned about the Ethical Approval.

Reply: Thank you for your comment. According to your advice, we insert the phrase “The study was conducted in accordance with the Declaration of Helsinki and approved by the Institutional Review Board of Hiroshima University Hospital (No.E-1518-1). Although the Ethics Committee of Hiroshima University Hospital waived the requirement for informed consent because we used anonymized data, informed consent was obtained using an opt-out option.” in the Material and Methods.

  1. Authors have not mentioned the Control(adjacent tissue from healthy control) for the Immunostaining and have not done the H&E staining.

Reply: Thank you for your comment. As you have mentioned, it is very important to consider the adjacent tissue of the donation target. Many prior reports have evaluated adjacent tissues of healthy controls. However, since our study focused on M carcinoma, T1 carcinoma, and advanced cancer lesions and examined TIL changes in tumor invasion, we were not able to consider adjacent tissue. However, this is a very important point and is therefore described in the limitation section.

  1. Retrospective nature of the study and use of single antibody bias needs to be elaborated.

Reply: Thank you for your comment. As you have mentioned, we believe that it is very important to prove that the current results can be reproduced in future prospective studies; thus, we have limited the study to a retrospective study. Regarding the antibodies, we used the one that showed the best staining performance as a result of using a variety of antibodies. However, we believe that it is very important to study using multiple antibodies in the future because of which we have included this information in the limitations section.

  1. English language and scientific language needs to be checked again.

Reply: Thank you for your comment. As you pointed out, we have resubmitted the document to an English editing company to have the text checked.

Reviewer 2 Report

Comments and Suggestions for Authors

The Authors analyzed the role of Tumor Infiltrating T cells in early stage CRC.  Sixty patients with CRC [20 patients each with intramucosal cancer (IM group), submucosal invasive cancer (SM group), and advanced cancer (AD group)] were randomly selected. We examined the changes in TILs with tumor invasion. Eighty-four patients with T1b CRC who underwent initial surgical resection with lymph node dissection or additional surgical resection with lymph node dissection after endoscopic resection were  selected. We examined the relationship between TILs and lymph node metastasis in these patients  with T1b CRC. The phenotype and number of TILs for all specimens were evaluated using the triple  immunofluorescence method for CD4, CD8, and Foxp3. All subtypes were predominantly more numerous according to the degree of CRC invasion. Furthermore, all subtypes were more abundant  at the invasive front of the tumor (IF) than in the center of the tumor (CT) in SM and AD groups.  The increased number of Foxp3 cells at the IF and the high ratio of Foxp3/CD4 and Foxp3/CD8 positively correlated with lymph node metastasis in T1b CRC. In conclusion, tumor invasion correlated with the number of TILs in CRC. The number and ratio of Foxp3 cells at the IF may help predict lymph node metastasis in T1b CRC.

QUESTIONS

1-Please specify the criteria of selection for the 60 specimen .

2-Please, specify the limits of triple  immune fluorescence staining . Despite the use of image analyzer, immune fluorescence allows a qualitative analysis .

3-Please, specify if the study was performed blindly or not. Were the researchers who analyzed immune fluorescence results blind to the clinical results, namely to lymph node involvement? Please, specify.

3-Pag 4 Line 140. The Authors found a significant difference in number of TIL between the three group of specimen according to the depth of  level of invasion.  In their analysis they considered this evidence as a prognostic factor. It is possible that this condition is the result of the depth of invasion of the tumor itself and it might represent a natural course of defense of the immune system. The increased number of Foxp3+T cell found with simultaneous lymph nodes metastases open several interpretations like

a)    The immune system is unable to stop tumor cells invasion . In this context Fox3+T cells may represent less effective cells or they may represent additional recruitment by the immune system of a “reserve group of immune cells”.

b)    I encourage the Authors to analyze in vitro the interaction  between Fox3+T cells and tumor cells for each individual patient. I guess that Fox3+T cells will be less effective than Fox3-T cells in neutralizing tumor cell invasion. Of course this may be the subject of a new research; however, this possibility should be introduced in the discussion. The basic concept I am going to support is that invasion of tumor cells is contrasted efficiently by the immune system in most patients. If the immune system is not efficient for genetic or acquired factors, we are going to see a significant immune reaction which testify at this  inefficiency of the immune system itself.

c)    In my opinion, lymph node involvement is an additional proof of the inefficiency of the immune system, which tries to recruit more T cells to contrast the tumor itself.

4- Fig 3 is the center of the results. I would like to suggest to divide in two figures.

CONCLUSIONS

Interesting study which suffers of the possibilities of biases related to the retrospective nature of the study itself. I encourage the Authors to perform a prospective study.

Comments on the Quality of English Language

I think that the English is acceptable. However, I am not English native and my opinion is not optimal.

Author Response

September 25, 2023

Dear Editor:

We thank you for the thorough review of our manuscript and your thoughtful suggestions and insights. We have taken the reviewer’s comments into account and have made adjustment to the text accordingly. We have highlighted changes in the manuscript in red.

The manuscript has been rechecked, and necessary changes have been made in accordance with the reviewers’ suggestions. The responses to all comments have been prepared and given below. I look forward to working with you and the reviewers to move this manuscript closer to publication in Cancers.

Sincerely,

Yasuhiko Kitadai, PhD

Department of Health Sciences, Faculty of Human Culture and Science

Prefectural University of Hiroshima

1-1-71, Ujinahigashi, Minami-ku, Hiroshima 734-8558, Japan

Responses to the Reviewer

Our point-by-point response to the comments is as follows.

  1. Please specify the criteria of selection for the 60 specimen

Reply: Thank you for your comment. Twenty patients with intramucosal CRC treated endoscopically between December 2011 and December 2015 at Hiroshima University Hospital were randomly selected. We also randomly selected 20 patients with T1b CRC who underwent surgical resection with lymph node dissection at the beginning or additional surgical resection with lymph node dissection after endoscopic resection during the same period. Finally, we randomly selected 20 patients with advanced CRC who underwent surgical resection with lymph node dissection during the same period. Thus, 60 cases were selected.

  1. 2. Please, specify the limits of tripleimmune fluorescence staining . Despite the use of image analyzer, immune fluorescence allows a qualitative analysis

Reply: Thank you for your comment. Triple immunofluorescence shows the localization of TILs from an overhead view; however, for counting using an image analyzer, it is necessary to increase the magnification to observe the same. To minimize the bias, images of the center of the tumor and invasive front of the tumor were taken at five random locations, and the number of TILs in each area was counted and averaged.

  1. 3. Please, specify if the study was performed blindly or not. Were the researchers who analyzed immune fluorescence results blind to the clinical results, namely to lymph node involvement? Please, specify.

Reply:

Thank you for your comment. The researcher analyzing the quantitative data of triple immunofluorescence was blinded concerning the data. However, another researcher subsequently integrated the quantitative data with the clinical data for evaluation.

  1. 4. 3-Pag 4 Line 140. The Authors found a significant difference in number of TIL between the three group of specimen according to the depth oflevel of invasion. In their analysis they considered this evidence as a prognostic factor. It is possible that this condition is the result of the depth of invasion of the tumor itself and it might represent a natural course of defense of the immune system. The increased number of Foxp3+T cell found with simultaneous lymph nodes metastases open several interpretations like
  2. a)The immune system is unable to stop tumor cells invasion . In this context Fox3+T cells may represent less effective cells or they may represent additional recruitment by the immune system of a “reserve group of immune cells”.
  3. b)I encourage the Authors to analyze in vitro the interaction  between Fox3+T cells and tumor cells for each individual patient. I guess that Fox3+T cells will be less effective than Fox3-T cells in neutralizing tumor cell invasion. Of course this may be the subject of a new research; however, this possibility should be introduced in the discussion. The basic concept I am going to support is that invasion of tumor cells is contrasted efficiently by the immune system in most patients. If the immune system is not efficient for genetic or acquired factors, we are going to see a significant immune reaction which testify at this  inefficiency of the immune system itself. 
  4. c)In my opinion, lymph node involvement is an additional proof of the inefficiency of the immune system, which tries to recruit more T cells to contrast the tumor itself.

Reply: Thank you for your valuable input. As you pointed out, the increase in the number of TILs in proportion to the degree of infiltration represents a natural defense process of the immune system. However, we also agree that the accumulation of inefficient Foxp3+ T cells may suppress tumor immunity and promote tumor progression.

In addition, high levels of lymphocyte aggregation and inefficiency of aggregating lymphocytes would be observed in congenitally or congenitally immunocompromised patients. It is very important to demonstrate this point, and as you point out, in vivo and in vitro studies would be very useful. In vivo studies would provide more insight into the inefficiency of TILs in cases of lymph node metastases. The important points underlying the interpretation of this study are discussed in the Discussion.

  1. 5. - Fig 3 is the center of the results. I would like to suggest to divide in two figures.

Reply: Thank you for your comment. According to your advice, we divided Figure 3 into two figures, Figures 3 and 4.

Reviewer 3 Report

Comments and Suggestions for Authors

Good methods and results with good clinical correlations.

Is the study numbers enough to reach the conclusion required.

Author Response

September 25, 2023

Dear Editor:

We thank you for the thorough review of our manuscript and your thoughtful suggestions and insights. We have taken the reviewer’s comments into account and have made adjustment to the text accordingly. We have highlighted changes in the manuscript in red.

The manuscript has been rechecked, and necessary changes have been made in accordance with the reviewers’ suggestions. The responses to all comments have been prepared and given below. I look forward to working with you and the reviewers to move this manuscript closer to publication in Cancers.

Sincerely,

Yasuhiko Kitadai, PhD

Department of Health Sciences, Faculty of Human Culture and Science

Prefectural University of Hiroshima

1-1-71, Ujinahigashi, Minami-ku, Hiroshima 734-8558, Japan

Responses to the Reviewer

Is the study numbers enough to reach the conclusion required.

Reply:

Thank you for your comment. As you pointed out, since this study was  a retrospective study conducted at a single-center, and the sample size was relatively small, we will have to conduct further studies to evaluate the validity of our conclusions.

Round 2

Reviewer 1 Report

Comments and Suggestions for Authors

The authors have almost answered all the previous comments. Due the retrospective nature of the study it has own limitations.

However, the manuscript can be considered publishable in the journal cancers.